# VECTORIZATION METHODS IN RECOMMENDER SYSTEM

## Abstract

The most used recommendation method is collaborative filtering, and the key part of collaborative filtering is to compute the similarity. The similarity based on co-occurrence of similar event is easy to implement and can be applied to almost all the situation. So when the word2vec model reach the state-of-art at a lower computation cost in NLP. An correspond model in recommender system item2vec is proposed and reach state-of-art in recommender system. It is easy to see that the position of user and item is interchangeable when their count size gap is not too much, we proposed a user2vec model and show its performance. The similarity based on co-occurrence information suffers from cold start, we proposed a content based similarity model based on doc2vec which is another technology in NLP.

## 1 INTRODUCTION

Recommender system is mainly used to provide item recommendation that the user may prefer (Ricci et al., 2015), and it can also be used to match candidate users to a particular item. Since the online shop, movie website have massive transaction and review data stored, it is possible to train the recommender system using these data. And there are some challenges like Netflix (Bell & Koren, 2007) which has effectively promote the process of recommender system . Now recommender system has been widely used in online shopping, books, music, movies, video games, news and social recommendation.

There are mainly three type recommendation techniques: collaborative filtering, content based and knowledge based (Lu et al., 2015). CF can be applied to almost all the recommendation situations, but it suffers from cold start problem. Both content based and knowledge based systems need domain knowledge like item attributes and user profile.

The progress in natural language processing and computer vision fields make us have more powerful tools to design and implement a recommender system. The word2vec method (Mikolov et al., 2013b) in NLP brings insights into recommender system. Barkan & Koenigstein (2016) proposed a item2vec model based on word2vec, and gains better result on the user-artist dataset. Due to the ability to deal with massive text and image information, the recommender system begin to deal with as more information as possible. The content2vec model (Nedelec et al., 2016) use multi module to process the text and image information into features.

In the paper we proposed two models. The first is user2vec model which is a mirror version of item2vec. And apply the user2vec model in the scenario where the user size is not too larger than item size. The second is a doc2vec model which convert the item description into vector, and use the vector to compute the similarity between items, then perform a item based collaborative filtering.

The rest of the paper is organized as follows: in section 2 we overviews the word2vec and doc2vec, in section 3 we present our methods, In Section 4 we describe the experiments setup and results, In Section 5 we summarize our findings.

## 2 RELATED WORK

This section describes the feature representation methods in NLP: word2vec and doc2vec. These two methods is very helpful in recommender system.

The bag-of-words model (BoW)(Harris, 1954) is the simplest way to extract features from text for machine learning. But it will loses information because it does not consider word ordering. Word2vec (Mikolov et al., 2013a) generate representation vectors out of words. And it is able to capture relations between words. In this paper we choose the word2vec model using Skip-Gram method(Mikolov et al., 2013b). It predict all surrounding words based on the word in center when training. And actually word is not limited to word, both item and user can be viewed as a word.

Doc2vec(Le & Mikolov, 2014) is a technique can convert the text of any length into a vector. It consider the paragraph vector as a sharing vector of all the words in the paragraph, and learn the paragraph vector by gradient descent. In recommender system, text information is very useful to learn the implicit attributes of item.

## 3 METHODS

In this section we describe the methodology for recommendation. We treat the recommendation on video games as a prediction task, based on the user id, item id, rating and item description information. The rating score has five values between 1 to 5. In user-based collaborative filtering, we predict based on the user neighbour's score on the particular item. In item-based collaborative filtering we predict based on the similar item's score given by the particular user.

### 3.1 USER2VEC

As a mirror version of item2vec Barkan & Koenigstein (2016), we treat user id as a word, and we place the users who rate the same item 4-5 score into a sentence called 'like sentence', and place the users who rate the same item 1-3 score into a sentence called 'dislike sentence'. We ignore the order of user id in the sentence by simply shuffle the sentence before training. And all the other user id in the sentence should be considered as the context for the current user id, So we set the window size to 9999999(a number larger than the max sentence size). After training, we can convert the user into a vector, and compute the compute the similarity between users by cosine. Then we perform a user-based collaborative filtering.

### 3.2 DOC2VEC

The description of books, movies and video games contains the implicit information which may match the preference of users. So the items are close in semantic vector space may have similar score. We train the doc2vec model by the corpus of the item descriptions in gensim and convert the item into vector by its description. Then we compute the similarity between items by cosine, and we perform a item-based collaborative filtering.

### 3.3 SCORE

When apply user2vec model we use user-based collaborative filtering, we calculate the score by the formula below

$$\hat{r}_{ui} = \mu_u + \frac{\sum_{v \in N_i^k(u)} sim(u,v) \cdot (r_{vi} - \mu_v)}{\sum_{v \in N_i^k(u)} |sim(u,v)|} \tag{1}$$

where $\hat{r}_{ui}$ is the prediction score of user $u$ to item $i$, $\mu_u$ is the average score given by user $u$, $N_i^k(u)$ is the $k$ nearest neighbour of user $u$, $sim(u,v)$ is the similarity between $u$ and $v$.

When apply doc2vec model we use item-based collaborative filtering, we calculate the score by the formula below.

$$\hat{r}_{ui} = \mu_i + \frac{\sum_{j \in N_u^k(i)} sim(i,j) \cdot (r_{uj} - \mu_j)}{\sum_{j \in N_u^k(i)} |sim(i,j)|} \tag{2}$$

## 4 EXPERIMENTS

### 4.1 DATASETS

The data for experiments is a subset of the amazon dataset on video games category (McAuley et al., 2015; He & McAuley, 2016), which has 231K review data originally. The review data includes user id, item id, rating and etc. The item metadata contain item id and description. After data processing, we remove the items without text description, and remove the users who has less than 5 records. Then we got 146K review data which includes 15K users and 9K items. These data are split into train data set(88K, 60%), validation data set(29K, 20%) and test data set(29K,20%).

### 4.2 MODEL AND PARAMETERS

For the purpose to compare between different models, we implement several models. First, we implement the item-based collaborative filter method as a baseline (Hug, 2017). It use similarity as KNN neighbourhood metric and score weight. In this model we set the max neighbour number k=40, and use cosine as similarity metric. Second, we implement a item2vec model Barkan & Koenigstein (2016) based on the word2vec method in gensim. It use skip-gram method, and set the negative sample number to 5, window size to 9999 which can include all the words in the sentence. Third, we implement a user2vec model which is a mirror version of item2vec. It's also based on word2vec . And we set the negative sample number to 5, window size to 9999999. Fourth, we implement a doc2vec model, it convert the item description into vector by using doc2vec method in gensim, and then perform a collaborative filter method based on the doc2vec similarity.

### 4.3 EXPERIMENT RESULTS

In table 1 the performance of knn-cf model is used as a baseline. It shows that item2vec model gains the best score on the test set and the performance of user2vec is slightly better than the baseline. It's demonstrated that the underlying word2vec model has effectively utilized the contextual information. And item2vec outperform user2vec is because the item frequency is higher than user frequency in reviews which is benefit to the training of a word2vec model. The doc2vec also outperform the knn-cf baseline may due to two reasons, one is the effectiveness of the doc2vec model, the other is the description text of video games has reflect the attributes of the item.

Table 1: Comparison between serveral methods

| Method | RMSE | MAE |
|--------|------|-----|
| knn-cf | 1.145 | 0.839 |
| item2vec | 1.135 | 0.849 |
| user2vec | 1.142 | 0.859 |
| doc2vec | 1.142 | 0.855 |

## 5 CONCLUSION

In this paper, we proposed two models: user2vec and doc2vec. And measure the performance on the video games subset of amazon. Our work enable us to transfer both user and item

into vectors. In future we plan to explore the relationship between the user vector space and item vector space.

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
