# OpenReview forum: "VECTORIZATION METHODS IN RECOMMENDER SYSTEM"
_ICLR.cc/2019/Conference_

### Official Review · AnonReviewer2 · 2018-10-12
**Paper seems quite preliminary and not ready for detailed review**

**Rating:** 3
**Confidence:** 4

**Review:**

The paper studies the use of embedding techniques in recommender systems, and shows that item2vec (an item vectorization method) can be replaced by user2vec, as users and items are interchangeable.

This is a reasonable enough idea, though not sufficient for publication in ICLR. I'd suggest the authors address the following details:
-- The methodological contribution is too small, and fairly obvious. Not sufficient for this conference.
-- Only evaluated on one dataset, so unclear whether the results are really representative
-- Comparisons against a very limited set of similar methods, which are probably not state-of-the-art for this dataset
-- The results don't seem significant, all methods compared perform almost equally

---

### Official Review · AnonReviewer3 · 2018-11-04
**(Short) Paper is too preliminary for this venue**

**Rating:** 2
**Confidence:** 5

**Review:**

The idea of learning user embeddings for downstream tasks in recommender systems is a good one.

However, this paper proposes no significant methodological developments (e.g., user2vec is an extension of item2vec obtained by transposing the observation matrix). Further, it does not present a thorough study with interesting empirical results (doc2vec does not improve performance, a single dataset is used, baselines are not state of the art).

Overall, this short paper (3 pages + refs) seems a bit preliminary and, in its current state, does not make a significant enough contribution to be accepted at this venue.

I would suggest that a more thorough analysis of similarity methods for NN models could be interesting to a recsys workshop or perhaps a conference focussed on recsys (e.g., ACM recsys).

---

### Official Review · AnonReviewer1 · 2018-11-07
**The paper is very drafty, the results preliminary and the basic idea not  sufficiently novel**

**Rating:** 2
**Confidence:** 5

**Review:**

Review:

— the writing is not sufficiently clear and a lot of the ideas are hard to follow (the sections 3.2 and 3.3 which should cover proposed methods are only a paragraph long each, have no loss functions and no architecture descriptions/diagrams)
— the ideas presented are only derivative and are not sufficiently novel for the venue
— the experimental section is incomplete having results on one dataset and not enough state-of-the art baselines. the uplfits look small and there is no discussion on statistical significance

---

### Meta-Review · Area_Chair1 · 2018-12-13
**Too preliminary for ICLR-2018**

**Confidence:** 5
**Recommendation:** Reject

**Metareview:**

The reviewers are unanonymous in their assessment that the paper is not ICLR quality in its current form.